

# Where is my arm? Investigating the link between complex regional pain syndrome and poor localisation of the affected limb

Valeria Bellan[1,2], Felicity A. Braithwaite[2], Erica M. Wilkinson[2], Tasha R. Stanton[2] and G. Lorimer Moseley[2]

[1] Cognitive and Systems Neuroscience Research Hub (CSN-RH), University of South Australia, Adelaide, South Australia, Australia
[2] IIMPACT in Health, University of South Australia, Adelaide, South Australia, Australia

Corresponding author
Valeria Bellan,
valeria.bellan@gmail.com

## ABSTRACT

**Background:** Anecdotally, people living with Complex Regional Pain Syndrome (CRPS) often report difficulties in localising their own affected limb when it is out of view. Experimental attempts to investigate this report have used explicit tasks and yielded varied results.

**Methods:** Here we used a limb localisation task that interrogates implicit mechanisms because we first induce a compelling illusion called the Disappearing Hand Trick (DHT). In the DHT, participants judge their hands to be close together when, in fact, they are far apart. Sixteen volunteers with unilateral upper limb CRPS (mean age 39 ± 12 years, four males), 15 volunteers with non-CRPS persistent hand pain ('pain controls'; mean age 58 ± 13 years, two males) and 29 pain-free volunteers ('pain-free controls'; mean age 36 ± 19 years, 10 males) performed a hand-localisation task after each of three conditions: the DHT illusion and two control conditions in which no illusion was performed. The conditions were repeated twice (one for each hand). We hypothesised that (1) participants with CRPS would perform worse at hand self-localisation than both the control samples; (2) participants with non-CRPS persistent hand pain would perform worse than pain-free controls; (3) participants in both persistent pain groups would perform worse with their affected hand than with their unaffected hand.

**Results:** Our first two hypotheses were not supported. Our third hypothesis was supported —when visually and proprioceptively encoded positions of the hands were incongruent (*i.e.* after the DHT), relocalisation performance was worse with the affected hand than it was with the unaffected hand. The similar results in hand localisation in the control and pain groups might suggest that, when implicit processes are required, people with CRPS' ability to localise their limb is preserved.

## INTRODUCTION

People with Complex Regional Pain Syndrome (CRPS) often report severe sensorimotor disturbances such as difficulty in localising their affected limb when they are not able to

directly look at it (*Lewis et al., 2007*, *2010*; *Lewis & McCabe, 2010*). Self-localisation is an important component of the self. For example, we know *this* body is *our* body because: (1) it feels like it is ours (sense of ownership); (2) each body part moves how we want it to move (sense of agency); (3) we know where each body part is (self-localisation) (*Serino et al., 2013*). Within different clinical conditions, such as CRPS, one or more of these components of self can be disrupted (*Moseley & Flor, 2012*). Here we are interested in self-localisation.

There is evidence that people with CRPS have impaired self-localisation during explicit tasks. For example, people with CRPS generally tend to be less accurate and slower than healthy controls when instructed to match the position of one limb to a bodily movement or position performed by the opposite limb (*Brun et al., 2019a*). While such findings support a generalised impairment, recent work suggests that CRPS may share features with spatial neglect, a neurological disorder arising after a lesion to the parietal cortex in which the person "neglects" the contralateral side of the world and/or of themselves, even in the absence of paralysis. For example, people suffering from neglect often show a sense of disownership (and sometimes even disgust) towards their affected limb, and this is associated with impaired self-localisation (see (*Caggiano & Jehkonen, 2018*) for a recent systematic review); similar features also occur in people with CRPS (*Lotze & Moseley, 2007*; *Moseley & Flor, 2012*). Recent work added to this picture by showing that in an implicit task involving a fake finger illusion (*Walsh et al., 2011*) people with unilateral upper limb CRPS appear less affected than controls, perhaps reflecting reduced weighting of cortical networks involved in bimanual hand tasks (*Wang et al., 2019*).

The idea of CRPS as a neglect-like condition, in which the *side of the space* usually occupied by the affected limb is affected—not just the *affected limb* itself—is supported by recent work. Reid and colleagues (*Reid et al., 2018*) found that people with unilateral upper limb CRPS were less accurate in drawing consecutive circles both with their affected hand, and with their unaffected hand generating the movement in the affected side of space (*i.e.* arm crossed over the midline). Other experimental and clinical studies highlight the potential involvement of the side of space in CRPS. In people with unilateral CRPS, temperature changes in *both* the affected and unaffected limb occur when the limb is placed in the affected side of space (*Moseley, Gallace & Spence, 2009* (although see (*De Paepe et al., 2020*) for conflicting results). Further, people with CRPS report a decrease in pain when the affected limb is positioned in the unaffected side of space, both physically (*Moseley, Gallace & Spence, 2009* and perceptually using prisms (see, for example, (*Bultitude & Rafal, 2010*; *Moseley et al., 2013*; *Christophe et al., 2016*).

Self-localisation of the body requires integration and weighting of inputs from various sensory sources (*e.g.*, the visual encoded position, proprioceptive encoded position), and is also influenced by space-specific encoding (*e.g.*, neglect-like involvement). Impairment of tactile (*Moseley & Flor, 2012*; *Stanton et al., 2013*; *Catley et al., 2014*) and proprioceptive (*Bowering et al., 2014*; *Stanton et al., 2016*) processing has been observed

in people with a range of persistent pain states, not just CRPS, and at present, it is not entirely clear what might underlie impaired self-localisation in people with CRPS or non-CRPS persistent pain. One way to delineate the influences of these features is to use sensory deception, whereby sensory input is manipulated to then determine the degree to which various inputs are relied upon to localise the limb. Such approaches can be considered to interrogate implicit mechanisms because the participant is naïve to the context of the task. A relevant sensory illusion that involves self-localisation of the hand is the Disappearing Hand trick (DHT, *Newport & Gilpin, 2011*). This illusion uses proprioceptive recalibration to induce visuo-proprioceptive incongruence, such that where you see the hand to be is not where it is actually located (*Bellan et al., 2015*, *2017*). The manipulation culminates when one hand is removed from view and the participant reaches over to touch the missing hand, realising then that it is not where they thought it to be. Our past work in healthy volunteers shows that even if the final step is omitted, re-weighting of proprioceptive and visual information still gradually occurs, such that over time, they begin to localise their missing hand as being closer to the proprioceptively encoded position than the visually encoded position. That is, closer to its true position (*Bellan et al., 2015*, *2017*).

The effects of illusions in people with persistent pain problems are difficult to predict. One report suggests that people with CRPS experience pain when faced with illusions that involve sensory incongruence (*Harris, 1999*; *Brun et al., 2019a*, *2019b*), including visual illusions such as the rabbit-duck effect (*Hall et al., 2011*). Other reports offer contrary findings—when sensory incongruence is induced *via* a rubber hand illusion (RHI; *Botvinick & Cohen, 1998*), they experienced the illusion similarly to healthy volunteers and did not report increased pain (*Reinersmann et al., 2012*, *2013*). Importantly, these studies that test the response of people with CRPS to illusions or sensory conflicts have involved explicit tasks. In other words, participants know that a conflict exists, or an illusion is occurring. For example, during the RHI (*Botvinick & Cohen, 1998*), participants are explicitly aware that the prosthetic arm is not their own arm. Unlike the RHI or doing paradoxical movements behind a mirror (*McCabe et al., 2005*), the DHT leaves participants genuinely naïve to what is actually happening (*Bellan et al., 2015*, *2017*). The potential importance of testing performance during implicit tasks is quite well established in studies with people with spatial neglect (for a review see *Berti, 2002*) but also, for example, with the case of cortical blindness, in which people 'see' nothing, but can correctly identify the colour of a shape presented in front of them (for a review see *Hadid & Lepore, 2017*).

Here we used the DHT to test the primary hypothesis that participants with CRPS would be less accurate than both pain-free controls and participants with non-CRPS hand pain in localising their hand after the DHT. Our secondary hypotheses were that (1) participants with non-CRPS persistent hand pain would be less accurate than pain-free controls in hand-localisation, and (2) that participants in both persistent hand pain groups would be less accurate in localising their affected hand than in localising their unaffected hand.

From herein we will consider the proprioceptively-encoded location as the 'true' location and deviation from that will be considered 'less accurate'. However, we accept that this definition serves the purpose of simplifying language while it relies on assumptions that cannot be validated.

## MATERIALS & METHODS

### Ethical approval

All participants gave written consent prior to participating in the study. The study was performed in accordance with the ethical standards laid down in the 1991 Declaration of Helsinki and was approved by the Human Research Ethics Committee of the University of South Australia (ID number 0000034649).

### Participants

Participants ($n = 60$) were divided into three groups: pain-free individuals ('Control'); individuals with a diagnosis of upper limb CRPS ('CRPS'); and individuals with pain in their upper limbs for more than three months, but who did not meet the criteria for a diagnosis of CRPS ('Non-CRPS Pain'). The Control group comprised 29 individuals (mean age 36 ± 19 years, 10 males), all right-handed except for one (ambidextrous). The CRPS group comprised 16 individuals (mean age 39 ± 12 years, four males); 12 were right-handed, three left-handed and one was ambidextrous. All CRPS participants had received a formal diagnosis by their health practitioner, however a further clinical assessment of signs and symptoms was performed before commencing the experimental session. The assessment was performed following the diagnostic criteria of the International Association for the Study of Pain (IASP) (*Harden et al., 2007*) (see Table S1). The Non-CRPS Pain group were 15 right-handed participants (mean age 58 ± 13 years, two males). For all of the participants in the CRPS and Non-CRPS Pain groups pain was located in the hand and, for most of them, the pain radiated to the entire upper limb and shoulder. A sample size calculation indicated a minimum total sample size of 36 ($n = 12$/group), in order to obtain a statistical power of at least 95% for medium effect size (f = 0.25) and an α set at 0.05. The eligibility criteria for each group is listed in Table 1.

### Procedure

Participants attended the Body in Mind lab in Adelaide, Australia, for a single one-and-a-half-hour session. All participants were asked to provide basic demographic data and to complete four questionnaires assessing their mood (Depression Anxiety Stress Scale-21, DASS-21; *Antony et al., 1998*; *Osman et al., 2012*), presence of pain catastrophizing thoughts (The Pain Catastrophizing Scale, PCS (*Sullivan, Bishop & Pivik, 1995*), and body perception disturbances (questions 1 to 6b of the Bath CRPS Body Perception Disturbance Scale, BPDS (*Lewis & McCabe, 2010*); question 7 was excluded because it could not be administered online, with minimal involvement of the experimenters). Participants belonging to the pain groups (CRPS and non-CRPS Pain) were asked to provide details concerning their condition (see Table S2).

**Table 1 Inclusion criteria.**

|  | Inclusion criteria |
| --- | --- |
| General (All Groups) | 1. Being over 18;<br>2. Ability to read and understand spoken or written English;<br>3. Absence of dermatological condition on the upper limbs (that might disrupt peripheral sensations);<br>4. Normal or corrected to normal vision;<br>5. No known psychiatric disorder[1] |
| Pain-Free Control Group | 1. Pain free with normal sensation in the arms and hands;<br>2. No current or past history of neurological issues (*e.g.* traumatic brain injuries, stroke, nerve injuries) or other medical conditions; |
| CRPS Group | 1. Diagnosis of upper limb CRPS[2];<br>2. No unresolved neurological or orthopaedic injury;<br>3. Pain elsewhere. |
| Non-CRPS Group | 1. Non-CRPS type persistent (lasting > 3 months) upper limb pain;<br>2. No unresolved neurological or orthopaedic injury;<br>3. Pain elsewhere. |

**Notes:**

[1] Except for Depression or Anxiety disorders, due to the large incidence of these conditions within the population living with persistent pain (*e.g. Vowles et al., 2020*).
[2] According to recommended criteria for research (*Bruehl et al., 1999*).

## Self-localisation

All participants performed a hand-localisation task in three different conditions, repeated twice (once for each hand, for a total of six trials), using the MIRAGE Multisensory Illusion Box (*Newport, Pearce & Preston, 2010*). Participants put their hands inside the box and, when looking down, they saw real time video of their hands that was either manipulated (in the DHT condition) or not (in the congruent conditions). Specifically, during the DHT condition, participants underwent the adaptation phase of an illusion called the Disappearing Hand Trick (DHT) (*Newport & Gilpin, 2011*). During this phase, unbeknownst to the participants, a mismatch is induced between the visually-encoded (*i.e.* where the participants see their hands to be) and the proprioceptively-encoded (*i.e.* where their hands really are) position of the hands. This leads to a localisation weighting that is biased towards the visual representation of the hand. Therefore, the participants see their hands moving inwards, while in fact, they have to move their hands outwards to keep the visual image of their hands central (see Manipulation Check below for more information).

During the two control conditions, the visually and proprioceptively encoded positions of the hand were congruent, such that the hands were exactly where the participants saw them to be. That is, during the Hidden Hand (HH) condition, the participants saw their hands moving outwards, and no manipulation was performed (*i.e.* the hands were actually moving outwards), while, during the Static Hand (SH) condition, the hands just hovered over the surface of the table inside the machine, staying centralised.

In all conditions, after 25 s (during which the participants either moved their hands or kept them still), they were asked to rest their hands on the bottom of the box and to keep looking at one of their hands, while it disappeared (a black square was superimposed over the image of their hand). Once the hand was out of view, a localisation task commenced: a red arrow appeared in the middle of the screen and started travelling at a constant speed towards the disappeared hand. The participants were asked to look at the side of the space where the hand was to be located and to keep track of where they felt their hand to be and say "stop" when they thought the arrow was pointing at their middle finger. For each trial, seven localisations were performed (one every 15 seconds). Then, both hands disappeared, and the actual location of the target hand was recorded as the baseline measure (see Fig. S1). Further details about the procedure and a video depicting the movement of the hands can be found elsewhere (*Bellan et al., 2015*, *2017*).

## Swelling and pain

At the beginning of the experimental session and after each trial, the experimenter measured the circumference of the participants' second and third fingers, and wrists of both hands, in order to assess any swelling change. In addition, participants belonging to the pain groups were asked to rate their pain on a scale between 0 (no pain at all) and 10 (worst pain imaginable).

## Manipulation check—disappearing hand trick

During the DHT (*Newport & Gilpin, 2011*), the adaptation phase explained above for the DHT condition was performed and, subsequently, after one hand disappeared, the participants were asked to reach across with their other hand to touch their disappeared hand. This "full DHT" (participants now aware that that their hand was not where they thought it was) was performed to verify whether the participants noticed any differences between the experimental and the control conditions. Specifically, if the illusion worked and the participants failed to touch their disappeared hand, they would be expected to show surprise and they were then asked whether they realised that their hands were not where they thought they were. All the participants were surprised and, when asked whether they noticed any difference between the conditions administered during the experiment, none of them were able to tell the difference between the control and the experimental conditions.

## Statistical analysis

Data were inspected and analysed by using custom-made scripts in *RStudio* version 1.2.1335 (*RStudio Team, 2020*) for Windows. The *lme4* package was used for linear mixed-effects models (LMM) (*Bates et al., 2015*). Different packages from the *Tidyverse* collection (*Wickham et al., 2019*) were used to explore and plot the data.

Akaike Information Criterion (AIC; *Akaike, 1974*), Bayesian Information Criterion (BIC; *Schwarz, 1978*) and log-likelihood were used to assess the fit of the LMM.

## Preliminary analysis

Data were visually inspected according to the two main hypotheses.

## Data handling

For each participant, localisation error was calculated as the difference (in pixels) between the actual position of the hand and the position of the hand as indicated by the participants at each point in time (*i.e.* where the participants stopped the arrow). We chose pixel as unity of measure because the error was measured directly on the screen where both the arrow and the hand appeared. The localisation error indicates the accuracy in localising one's own limb, with higher scores suggesting a larger discrepancy between the actual and perceived position of the limb, and, therefore, lower accuracy.

## Primary hypothesis and secondary hypothesis 1: LMM1

According to the primary hypothesis, participants in the CRPS group would show larger localisation error ('Error') than Controls and the Non-CRPS Pain group; at the same time, according to the first secondary hypothesis, participants in the Non-CRPS Pain group would in turn show larger localisation error ('Error') than Controls. The effects of 'Time' ($T_1$ to $T_7$) and 'Condition' (DHT, Hidden, Static) on 'Error' were plotted for each participant. High interindividual variability meant that the factor 'Participants' was included in the final model as random factor, with 'Condition', 'Time' as fixed effects, and 'Error' as the dependent variable (see Fig. S2). In line with previous results (*Bellan et al., 2015*, *2017*) we expect a main effect of Time for all groups.

## Secondary hypothesis 2 and exploratory analysis: LMM2

According to the Secondary hypothesis 2, participants with hand pain would show larger localisation error when judging location of their affected than their unaffected limb. The difference in the average localisation error over T1–T7 between limbs performed by both groups for each condition was plotted. Because a difference was found to exist between the localisation of the affected and unaffected hand within the DHT condition, the decision was made to include the factor Hand (affected, unaffected) in the second model.

Hand laterality was considered as a covariate. In this regard, previous research (*Grabherr et al., 2019*) showed no difference in localisation error between the right and the left hand, in right-handed participants. The current study involved two clinical groups with hand pain, so an effect of handedness (right, left) would have been difficult to interpret, because either the right or the left hand could have also been the affected. Therefore, in order to still account for the effect of handedness, the factor 'Hand' (affected, unaffected) was only included in the second LMM, in which only participants in the CRPS and non-CRPS Pain groups were included.

Finally, in the pain groups separately, Error was plotted against: Age, Sex, change in Swelling and Pain ratings (both calculated as the difference between baseline and after each condition), DASS scores, PCS score, BPDS scores, and Duration of the painful condition (in months). Because no specific hypotheses were made for these comparisons, after visual inspection of the data, the factors 'Age' (years) and 'Anxiety' (the score of anxiety from the DASS questionnaire) were selected to be included in the second LMM.

## RESULTS

### Localisation error

*Effect of group on localisation error (Primary Hypothesis and Secondary Hypothesis 1)*

All significant effects and interactions of the LMM as well as the model fit of localisation error are included in Table S3.

LMM results indicated a main effect of Condition ($F_{(2,2460)}$ = 1040.326, $p < 0.001$), revealing that participants were significantly more inaccurate (larger localisation error) in the DHT condition (M = 100.71, SD = 39.67 pixels) than in the two control conditions (*i.e.* Hidden (M = 20.95, SD = 22.05 pixels) and Static (M = 20.78, SD = 17.26 pixels)), regardless of group. In addition, participants' accuracy varied significantly over consecutive localisations, as indicated by a main effect of Time ($F_{(1,2460)}$ = 22.56, $p < 0.001$). Specifically, significant interactions between Time and Condition ($p < 0.001$) revealed that participants became more accurate over time during the DHT condition but not during the control conditions, which reflected the large initial error in the DHT condition but not in the control conditions (see Fig. 1).

Finally, there was no effect of Group ($p = 0.54$), suggesting that, overall, CRPS participants were not less accurate than the other two groups, which was contrary to our prediction (see Fig. 2). However, a significant interaction between Group (non-CRPS Pain) and Time ($p < 0.001$), and between Group, Time and Condition ($p = 0.002$) suggested that the localisation error in the DHT condition decreased more rapidly in the non-CRPS Pain group than it did in the other two groups (see Fig. 1).

*Influence of affected hand on localisation error in those with persistent hand pain (secondary hypothesis 2)*

All significant effects and interactions of the LMM as well as the model fit of localisation error are included in Table S4.

The LMM showed a main effect of Condition ($F_{(2,1271)}$ = 8.62, $p < 0.001$), replicating the results reported above, with the largest error in the DHT condition (M = 94.8, SD = 40.38 px; Hidden: M = 21.68, SD = 25.48 px; Static: M = 23.45, SD = 18.9 px). A significant interaction between Condition and Group ($p = 0.02$) also suggested that the non-CRPS Pain group showed a significantly smaller difference between DHT and the control conditions than the CRPS group did (see Table 2).

Furthermore, a second significant interaction between Affected Hand and DHT Condition ($p = 0.03$) suggested larger error for the affected hand than the unaffected hand during the DHT condition than during the control conditions (Fig. 3, Table 3).

### Exploratory analyses

A significant interaction between DHT Condition and Age ($p < 0.001$) showed that older age was correlated with smaller error in the DHT condition (the older the age, the smaller the error), while the opposite was true for the Static and Hidden conditions (Fig. 4A). The same result was also shown in the significant interaction between Group, Condition and Age ($p = 0.007$), whereby the non-CRPS Pain group showed a stronger difference

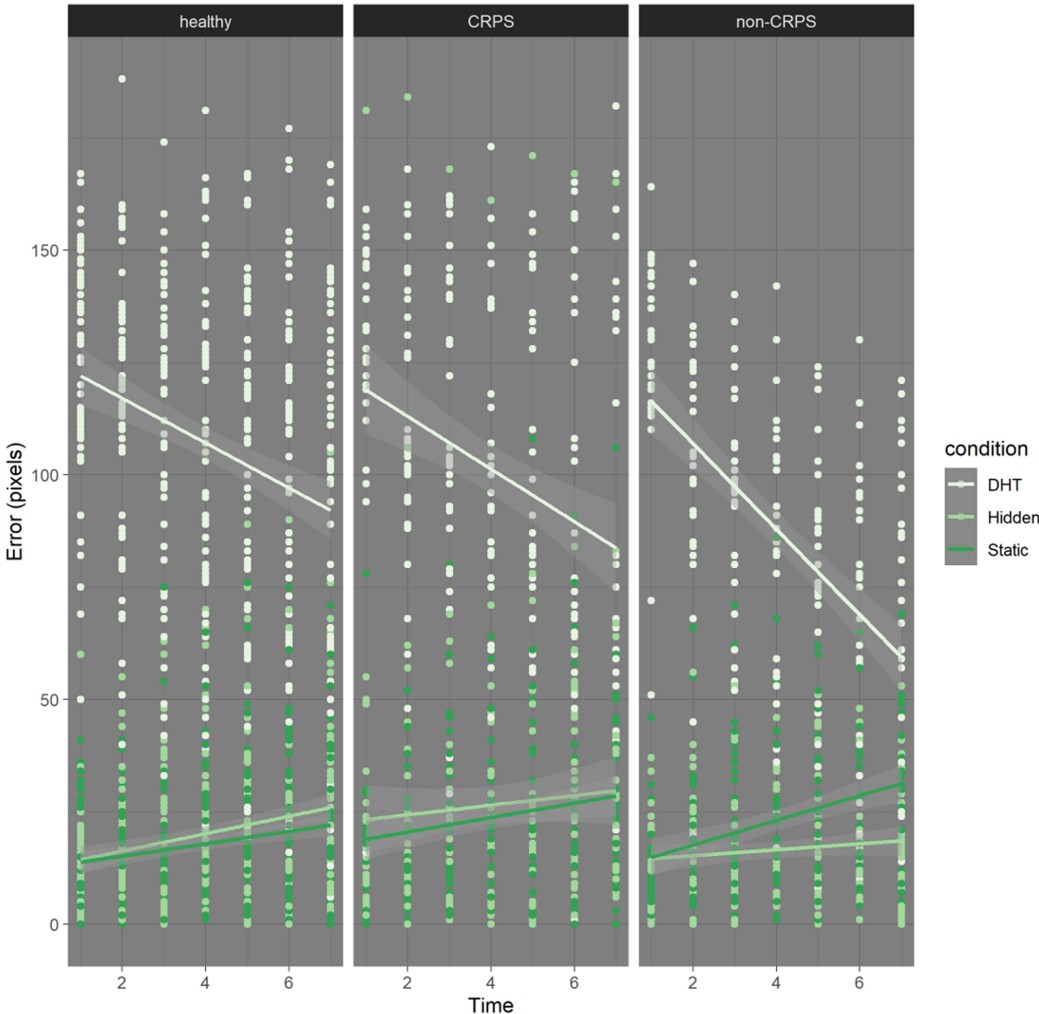

**Figure 1 Error by time by condition by group.** The dots represent each localisation made by the each participant in each group. In the DHT condition (in very pale green) all participants tend to start with a larger error compared to the other two conditions, but they become more accurate by time, even though the error never reaches 0 (*i.e.* no participants was able to correctly localise their hand). Also, the steepness of the red lines support the idea that participants in the Non-CRPS Pain group were probably faster in their reliance on proprioception, as shown by a steeper line towards 0, with an initial intercept similar to the other two groups.               

between the two control conditions and the DHT condition in the same direction (with positive correlation between Age and Error in the control conditions, and a negative correlation for the DHT condition) than the CRPS group did (Fig. 4B) (although note that participants in the Non-CRPS Pain group were older than participants in the CRPS group). Finally, the same significant trend ($p = 0.02$) was found in the comparison between the affected and unaffected hands, showing that the affected hand showed a stronger difference than the unaffected hand between conditions in the same direction (where a positive correlation was found between Age and Error in the control conditions, and a negative correlation for the DHT condition) (Fig. 4C).

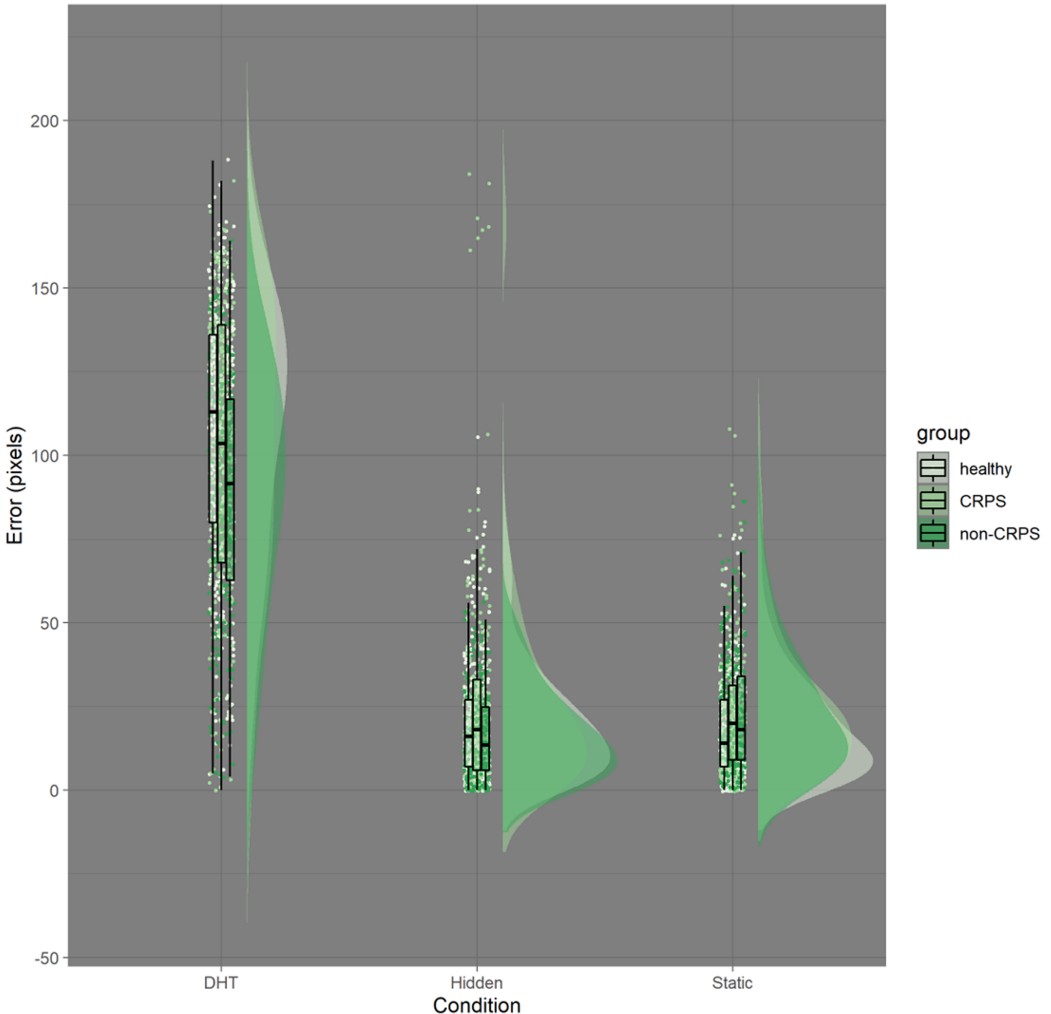

**Figure 2 Error by condition by group.** The raincloud figure represents the distribution of the error (y axis) for each participant and for each condition (x axis). The groups are colour coded. Even though the distribution is quite similar across different group, the CRPS group seems to have a larger variability across participants (*e.g.* each participant seems to behave quite differently compared to the others within the same group). However, contrary to our prediction, the participants in the CRPS group were not less accurate overall compared to the other groups.

**Table 2 Localisation error: CRPS Pain and Non-CRPS Pain groups.**

| | Pain groups (M ± SD) (px) | |
| --- | --- | --- |
| | **CRPS** | **Non-CRPS** |
| DHT | 101.26 ± 43.29 | 87.9 ± 34.84 |
| Hidden | 26.48 ± 32.28 | 16.56 ± 13.48 |
| Static | 23.74 ± 19.65 | 23.12 ± 18.09 |

Note:
Localisation error (in pixels) for the two pain groups across the three different conditions.

Our exploratory analyses showed a significant interaction between Group, Condition and Anxiety ($p < 0.001$) such that in the Non-CRPS Pain group higher anxiety was significantly and positively correlated with larger error only in the DHT condition

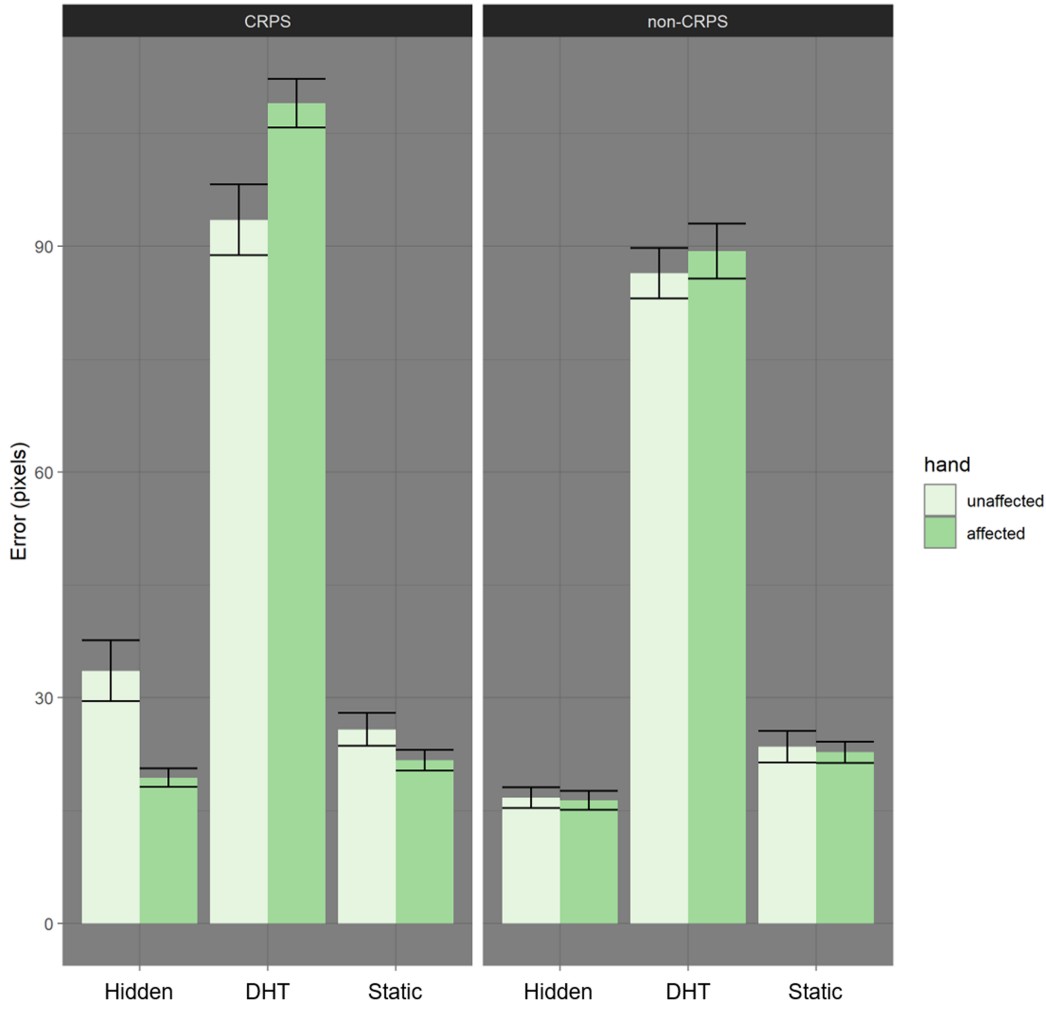

**Figure 3 Error by condition by hand.** Both groups show larger localisation errors for the DHT condition compared to the other two control conditions. However the difference between affected and unaffected hand is smaller in the non-CRPS than in the CRPS group.

**Table 3 Localisation error: affected and unaffected hand.**

| | Hand (M ± SD) (px) | |
| --- | --- | --- |
| | **Affected** | **Unaffected** |
| DHT | 99.51 ± 37.02 | 90.09 ± 43.05 |
| Hidden | 17.93 ± 13.04 | 25.44 ± 33.21 |
| Static | 22.21 ± 14.56 | 24.68 ± 22.37 |

Note:
   Localisation error (in pixels) for the two pain groups across the three different conditions comparing the affected and the unaffected hand.

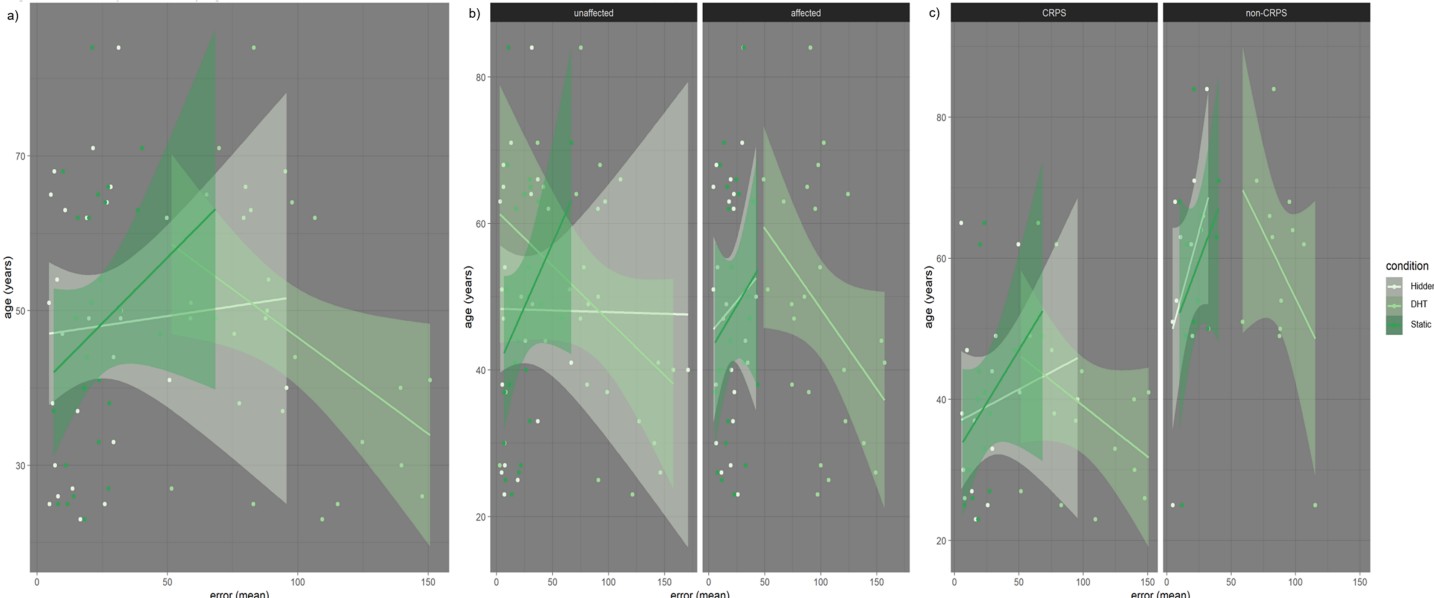

**Figure 4 Error by Age.** (A) Older age seems to be correlated with smaller error but only for the DHT condition, while the opposite trend can be observed for the other two conditions. This appears particularly clear when the two groups are considered separately (B), especially as far as the Non-CRPS Pain group is concerned. However, (B) also shows that, in general, Non-CRPS Pain group participants were older than the participants in the CRPS group. Interestingly enough, though, by comparing the affected and unaffected hand (C) (instead of the two groups) the effect is still present, but the difference between the DHT and the two control conditions is stronger for the affected compared to the unaffected hand.

($p = 0.04$), with the two control conditions showing the opposite trend (*i.e.* the higher the anxiety level, the smaller the error). In the CRPS group, higher anxiety was positively correlated with larger error for all three conditions, however this correlation was not significant (Fig. 5).

## DISCUSSION

This study evaluated hand-localisation accuracy in people living with upper limb CRPS, by comparing their performance to that of a clinical control group (people living with non-CRPS persistent hand pain), and with a non-clinical control group (pain-free individuals). Our hypotheses that participants with CRPS would be less accurate in hand-localisation than pain-free controls and participants with pain of other origins, and that all participants with persistent hand pain would be less accurate than pain-free controls were not supported. With regards to our Secondary hypothesis 1, we found the opposite of our prediction—people living with non-CRPS hand pain showed a quicker re-weighting toward proprioception after the DHT than the other two groups did - possibly indicating a smaller susceptibility to the illusion. Our Secondary hypothesis 2 was partially supported—participants living with hand pain were less accurate in localising their affected than their unaffected hand, but only in the DHT condition, involving incongruent proprioceptive and visual encoded representations. Crucially, no significant difference between the groups or between the affected and unaffected hands was found for the control conditions, suggesting that, under normal circumstances (when accurate

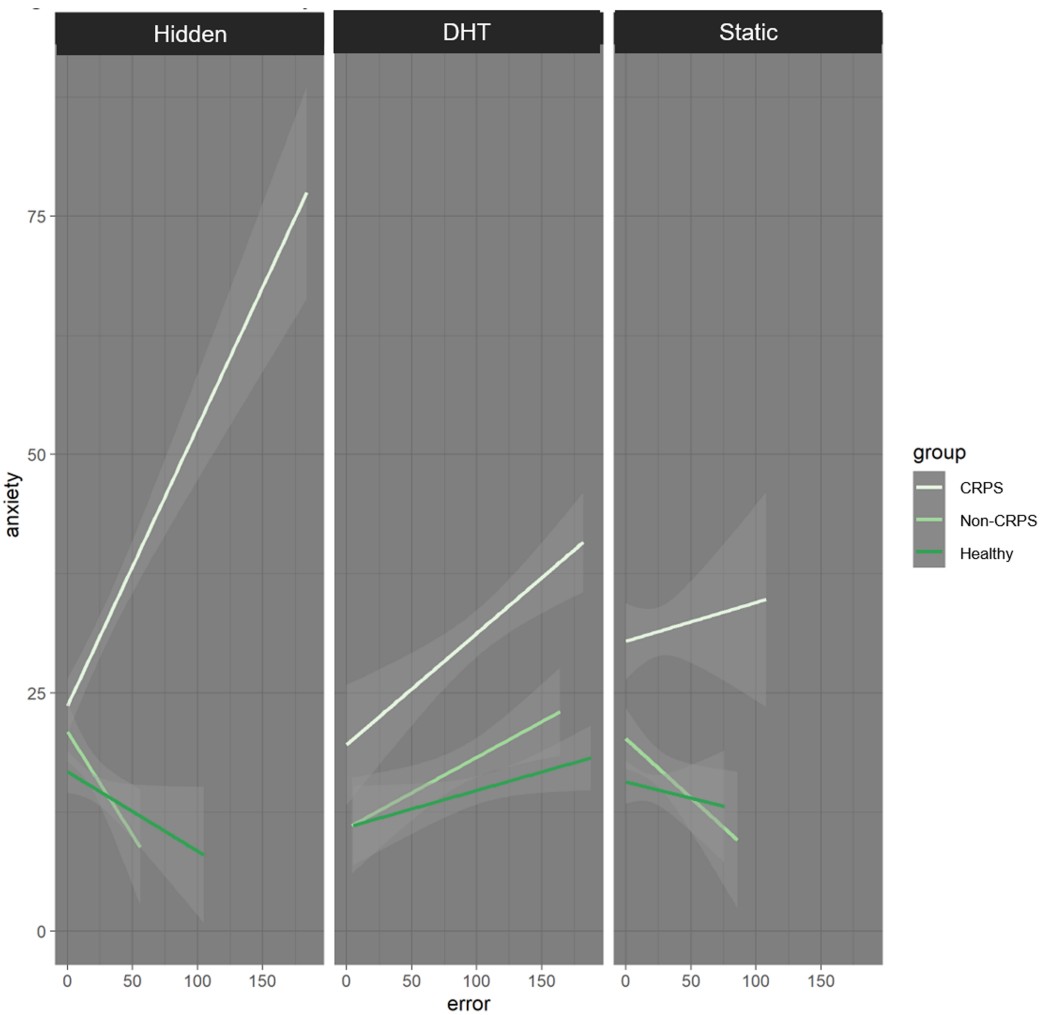

**Figure 5 Anxiety by Error.** The CRPS group (very pale green line) shows the same positive correlation between anxiety and error across all conditions. The Non-CRPS Pain (pale green line) and the Healthy (dark green line) groups show the same positive correlation but only for the DHT condition (and the opposite for the control conditions).

visual and proprioceptive cues are provided) persistent pain, whether in association with CRPS or not, does not appear to affect the self-localisation accuracy of the affected body part.

People with CRPS tend to report difficulties in localising their own affected limb when they cannot see it. This has been anecdotally reported and experimentally studied under different conditions (*Lewis & McCabe, 2010*; *Lewis et al., 2010*; *Reinersmann et al., 2012*). Several studies found that these people are less accurate than those without CRPS in tasks in which an accurate spatial representation of the limb is required (*e.g. Reid et al., 2018*; *Brun et al., 2019a*, *2019b*). A range of other findings point to a kind of spatial (*Galer & Jensen, 1999*), or so-called 'somatospatial' neglect (*Reid et al., 2016*). However, this idea remains controversial because the total picture is confusing. For example, a very recent study (*De Paepe et al., 2020*) failed to replicate previous studies that indicated

the presence of a perceptual bias away from the affected limb during a temporal order judgement task in CRPS participants (see also *Bultitude, Walker & Spence, 2017*). The present results add weight to the argument against neglect-like aspects of CRPS because, if proprioceptive encoded location had less weighting in CRPS, then the error during the DHT condition would have been greater in the CRPS group than it was in the other groups, which was not the case.

Finally, and crucial for the present study, vast literature has now shown the dissociation between implicit and explicit processing of spatial representation in studies involving people with neglect. For example, people with neglect show a dissociation in accessing the mental number line: they fail to access it explicitly, but they are accurate if asked to do so *via* an implicit task by using an alternative strategy that does not involve the damaged brain areas (*Priftis et al., 2006*). Even if the neglect-like metaphor stands true for people with CRPS, this possible dissociation needs to be considered. The peculiarity of the illusion employed in this study is that participants rarely realise they have been tricked and, if they do realise, it only happens after they fail in an attempt to physically locate their hand using their other hand. This means that, at least in part, the localisation task (which precedes the physical attempt to touch the disappeared hand) is conducted under the assumption that the hand is where the participant last saw it. Therefore, the participants are led to believe they are doing an explicit localisation task, while, in fact, they are not. Future studies will need to further investigate the possibility of a dissociation between implicit and explicit tasks by possibly using paradigms employed in the study of people with neglect.

Interpretation of the current findings should consider the likely underpinnings of the illusion we used. The DHT relies on visually encoded data having greater weighting than proprioceptively encoded data, and thus the finally perceived location of the hand reflects the former more closely than the latter. That the error was greater for the affected hand than it was for the unaffected hand for both persistent hand pain groups (*i.e.* the illusion was 'stronger') implies that the extent to which visually encoded data outweighs proprioceptively encoded data is greater for the affected than the unaffected hand in these groups. This would be predicted on the basis of use-dependency of neural networks: using the affected hand less than the unaffected one could explain our results, and might be mediated by impaired proprioception. Relevant here is a study in people with CRPS that aimed to interrogate the relative weighting of bimanual cortical representations (*i.e.* neural networks that underpin two-handed tasks) and unimanual cortical representations (*i.e.* those that underpin one-handed tasks), through a procedure that induces the illusion that one's index fingers are closer to each other than they really are (*Walsh et al., 2011*; *Wang et al., 2019*). That study showed a smaller illusion in people with CRPS than in healthy controls, which implies that the extent to which bimanual cortical representations outweigh unimanual cortical representations is less in people with CRPS than it is in healthy controls. Again, this result would be predicted on the basis of use-dependency of neural networks: avoiding bimanual tasks could explain that result. That people with CRPS perform normally on the RHI (*Botvinick & Cohen, 1998*; *Reinersmann et al., 2013*) does not contradict our evidence of altered response on the

localisation task, because the RHI involves a different mechanism - synchronous visual and tactile input allocated by the brain to the same event.

The current results raise interesting reflections on why people with CRPS-related persistent pain in particular can report 'losing track of their hand' during daily living. Perhaps our results offer a potential explanation. That is, without using the hand, or indeed even looking at it, prevents the usual opportunities for brain-held models of the body and its location to be updated. That patients report that seeing their hand surprises them because they thought it was somewhere else, not because they thought it was missing, points to this failure to update. Although speculative, this scenario shares parallels with asomatognosia, where brain damage affecting proprioceptive or motor function, combined with not looking at a hand, can lead to a failure to recognise a limb or part thereof (*Mendoza, 2011*).

Older people were both less accurate during control tasks (when visually encoded data were not manipulated) and more accurate during the DHT. This interesting finding also raises an important caution because it opens the possibility that age of participants may have contributed to the non-CRPS hand pain group showing a quicker relocalisation than the other groups. Why might age affect performance in this way? Perhaps proprioceptively encoded models of the body continue to improve with age. This would mean that, when vision is removed, older people update their internal model of body position more quickly, as we saw here. However, this would also result in more accurate performance in control tasks, which we did not observe. Alternatively, age is known to be associated with decreasing reliability of visually encoded data (*Kolesnikov et al., 2010*), which would explain less accuracy during control tasks, smaller illusion effect during the DHT, and more rapid recovery after the DHT. In addition, *Laurienti et al. (2006)* reported that older people showed enhanced multisensory integration: this population show a better performance when there are two stimuli providing the same information than when a single stimulus is providing it. The authors of that paper linked this finding to an offset decline in signal-to-noise-ratio with increasing age. It is thus interesting to consider that this enhanced integration might also involve enhanced detection of multisensory incongruence. Clearly, more research is required to understand this interesting finding.

Finally, exploratory analysis seems to suggest that our results cannot be explained by sex or mood. Although accuracy tended to be worse in those who were more anxious according to the DASS-21, there seemed to be no specific effect on the outcome of the DHT. Another visuotactile illusion in which visual input of touch to the affected hand causes pain in people with CRPS (*Acerra & Moseley, 2005*) but not people with non-CRPS neuropathic pain (*Krämer et al., 2008*), and other disruptions of bodily awareness are thought to be related to levels of distress (*Breimhorst et al., 2018*). That finding might imply a role for mood in problems integrating bodily data in CRPS, at least in what might be called 'explicit' problems of bodily awareness. This study, alongside that of *Wang et al. (2019)*, early studies involving implicit motor imagery (*Schwoebel, Boronat & Branch Coslett, 2002*, *Moseley, 2004*), and a finding of a lower heartbeat-evoked potential

amplitude in people with CRPS than in healthy controls (*Solcà et al., 2020*) all involve implicit mechanisms, not explicit.

Interpretation of the current work should consider its limitations. First, we did not lodge and lock our protocol and statistical analysis plan prior to data collection. When we commenced this study, such practice was uncommon in our field, but now it is recommended, and our group is among those at the forefront of this push (*Lee et al., 2018*). Failure to do this clearly represents a shortcoming in transparency and reporting. Other limitations include the age disparity between the groups, with the Non-CRPS Pain group being on average older than the other two groups. In addition, the participants in that group reported a much longer duration of the painful condition. The amount of time spent being in pain might have an effect on physiological (*e.g.* cortical reorganization) as well psychological processes (*e.g.* level of distress). Finally, we interpret the results as reflecting changes in the relative weighting of visual and proprioceptive-encoded data, but we did not explicitly assess either.

## CONCLUSIONS

In conclusion, contrary to our hypotheses, being in pain does not seem to necessarily lead to worse self-localisation abilities. However, possibly due to use-dependent effects, people tend to perform better with the unaffected hand compared to the affected counterpart. Future studies will need to clarify whether this disadvantage of the affected hand is body-centered or body part-centered. Interestingly, the similar performance in localisation abilities of people with CRPS and pain-free controls led to the interpretation of a possible dissociation between implicit and explicit neural processes in CRPS. This would once again suggest the existence of neglect-like characteristics in CRPS.

## ACKNOWLEDGEMENTS

The authors would like to thank Ms. Niki Najafi and Ms. Brooke Wilkin for their help during testing procedures. In addition, we would like to thank all the participants that volunteered their time to take part in this study. Finally, a special thanks to the Australian CRPS community.

### Funding

Valeria Bellan has received funding from the University of South Australia ("Bottleneck Operation") to cover part of the costs of this publication. Tasha R. Stanton is supported by a National Health & Medical Research Council (NHMRC) of Australia Career Development Fellowship (ID1141735). G. Lorimer Moseley was supported by a Leadership Investigator Grant from the NHMRC of Australia ID1178444. The funders had no role in study design, data collection and analysis, decision to publish, or preparation of the manuscript.

## Grant Disclosures

The following grant information was disclosed by the authors:
University of South Australia ("Bottleneck Operation").
National Health & Medical Research Council (NHMRC) of Australia Career Development: ID1141735.
Leadership Investigator Grant from the NHMRC of Australia: ID1178444.
Australia, Europe and North America, AIA Australia, the International Olympic Committee, Port Adelaide Football Club, Arsenal Football Club.
NOIgroup (Neuro Orthopeadic Institute).
Dancing Giraffe Press & OPTP.

## Competing Interests

G. Lorimer Moseley is an Academic Editor for PeerJ. He has been reimbursed by professional and scientific bodies for travel costs related to presentation of research on pain at scientific conferences/symposia, and has received support from: Reality Health, Connect Health UK, Seqirus, Kaiser Permanente, Workers' Compensation Boards in Australia, Europe and North America, AIA Australia, the International Olympic Committee, Port Adelaide Football Club, Arsenal Football Club. G. Lorimer Moseley has also received speaker fees for lectures on pain and rehabilitation, and receives book royalties from NOI group (Neuro Orthopeadic Institute) publications, Dancing Giraffe Press & OPTP for books on pain and rehabilitation. Tasha R. Stanton has received speaker fees related to presentations on pain and rehabilitation.

## Author Contributions

- Valeria Bellan conceived and designed the experiments, performed the experiments, analyzed the data, prepared figures and/or tables, authored or reviewed drafts of the paper, and approved the final draft.
- Felicity A. Braithwaite performed the experiments, authored or reviewed drafts of the paper, and approved the final draft.
- Erica M. Wilkinson analyzed the data, prepared figures and/or tables, authored or reviewed drafts of the paper, and approved the final draft.
- Tasha R. Stanton conceived and designed the experiments, authored or reviewed drafts of the paper, and approved the final draft.
- G. Lorimer Moseley conceived and designed the experiments, authored or reviewed drafts of the paper, and approved the final draft.

## Human Ethics

The following information was supplied relating to ethical approvals (*i.e.*, approving body and any reference numbers):

The study was approved by the Human Research Ethics Committee of the University of South Australia (ID number 0000034649).

## Data Availability
The raw measurements are available in the Supplemental Files. The raw data shows localisation errors, scores at psychological assessments and duration of the pain conditions (where appropriate) for all groups.

## Supplemental Information
Supplemental information for this article can be found online at http://dx.doi.org/10.7717/peerj.11882#supplemental-information.

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
