# Peer review of "Where is my arm? Investigating the link between complex regional pain syndrome and poor localisation of the affected limb"

_PeerJ, doi:10.7717/peerj.11882_

## Round 0.1 · original submission · Minor Revisions

I apologize for the long delay that you had to suffer from the time of submission of your manuscript: it was not easy to secure expert reviewers, as recently happens rather often. However, I am back with a decision which stems from the two reviews - both suggesting that the manuscript has larger merits than problems and indicating "minor revisions". This is one case in which the combination of two "minor revisions" might have better been translated into a major revision, but I wanted to keep the overall positive attitude of both reviewers because 1) I also find that your work is very interesting and deserves publication and 2) the reviewers' feedback can be handled with interventions on text, graphs and partly on revising the way some results are obtained/interpreted. At any rate, I invite you to respond to all points raised by the reviewers in a revised version of the ms and a detailed point by point reply letter. I am looking forward to receiving the revised ms and wish you all the best.

Reviewer 1 ·

Basic reporting

This is an interesting study on an important topic. There are not many studies investigating self localization in chronic pain patients. Especially the difference between the affected and the unaffected had side as found here (hypothesis 3) is very interesting. But also the impact of the factors age and anxiety on error performance are important findings.
The language is correct-the writing very good. The Intro & background to provide a comprehensive overview about the context and the literature is well referenced & relevant. The Structure conforms to the standards. The Figures are relevant, high quality, well labeled & described. It might be interesting for people not knowing the DHT paradigm to include a methods figure illustrating this paradigm. As far as I could overview there were no raw data included. However, the data are plotted very clearly in the figures and additional raw data might not be necessary here.
Table 2 and Table 3 seem to be completely taken from the stat. software. I think it should be presented more user friendly. In addition please describe stat. values also in the Results section. It is quite difficult for the reader to find the stat. results in the table when not explicitly provided in the text.

Experimental design

The original primary research is within the scope of the journal. The research question well defined, relevant and meaningful. The investigation was performed to a high technical and ethical standard. Overall, methods described with sufficient detail and information to replicate. The statistical comparisons are sound and described correctly.

Validity of the findings

The findings are highly informative and new. The assumptions made from these findings are justified. Replication of the paradigm possible with the information provided. Conclusions justified.

Additional comments

The following information might be provided:
A. Illustration of the experiment in a methods figure.
B. Improve readability of Table 2 and 3
C. Insert stat. values in the Results section.
D. Discussion (page 16, top): It is not comprehensible that the authors discuss distortion of somatosensory finger representation when addressing body matrix changes in CRPS. It would be more appropriate to address findings on the (right) intraparietal sulcus here.
Also on page 17: “…visually encoded data outweighs proprioceptively encoded data is greater for the affected than the unaffected hand in these groups…” The increased effect of the illusion in CRPS patients might well be related to proprioceptive impairment then? The same in elderly: decrease in visual function decreases then DHT effect since it is based on ten visual domain.
Might be also an issue for the limitations: visual and proprioceptive performance not explicitly tested here.
Conclusion: not very creative- just a mere repletion of main results.
Minor:
Citation error line 366: “First, Di Pietro et al (CITE)’s meta- 
...“

Reviewer 2 ·

Basic reporting

All good.

Experimental design

Some suggested improvement to analysis made in attached document. Otherwise, all good.

Validity of the findings

No comment (all good).

Additional comments

This ms describes using creating incongruence between seen and felt hand positions in order to test the implicit localisation of the hand in CRPS alongside non-CRPS and non-pain controls. The results do not bear out a particular localisation deficit in CRPS, but do suggest that the affected hand is not localised as accurately in CRPS and non-CRPS pain. The ms is well-written, although the authors fall into the trap slightly of not being able to put themselves in the shoes of someone who has never seen the DHT. Overall, this is publishable, but there are (as always) a number of points to consider/address. Most of these are entirely unproblematic, some require further thought, explanation or discussion.

Potentially problematic:

The introduction is clear, but the hypotheses are missing a step from the introduction. There is no discussion in the introduction around limb localisation in people with non-CRPS chronic pain. This is required in order to formulate the hypotheses involving the non-CRPS pain group. If no data exists, the this becomes more exploratory than hypothesis driven.

It is not clear whether error was absolute or directional. Which was it and why? It would seem from Figure 1, and supplementary figures, that error was coded as absolute, but is this fair? Presumably, in the static and hidden hand conditions, errors could be slightly to the left (say, -ve) and right (+ve) of the real hand, but in the DHT, the error could only ever be +ve (towards the visual representation (is that right?). If you do that, then you lose the characterisation of noise around the judgment that is made absolute in the Static and Hidden conditions, but remains directional in the DHT measurements. The upshot of this is that you reduce the variability in the static condition and not in the DHT. It might also disguise directional errors in your CRPS group that you might expect to see if you subscribed to the spatial neglect hypothesis.


Points needing further discussion:

The argument around usage and weighting is interesting. The results here suggest that proprioception itself is not the problem, but that it may be given less weighting is. That could help to explain why CRPS patients ‘lose track’ of where their hand is in daily living. Not using the hand fails to update representations and proprioception is (sort of) ignored. CRPS patients tend not to like looking at their affected hand either. Thus, without using the hand, looking at it or integrating proprioception from it effectively – how are you to know where it is? You only know by looking at it and they don’t do that if they can help it. I can imagine the mechanisms being similar to asomatognosia (which the DHT was designed to replicate according to the original) – failure to look towards the hand plus lack of proprioception and lack of motor updating = missing hand. Just a thought.

Is it interesting that pain ratings did not affect errors? Does this suggest that it is not pain per se that disrupts body representations, but the condition itself? Maybe not, if the affected hand is mislocalised more in both pain groups.

I am intrigued by the anxiety observations (more so than the authors, I suspect). If I have interpreted the results correctly, raised anxiety seemed to lead to greater errors. Why would this be? I think that the authors do not consider this point enough. Could increased errors, when anxious, be a result of even greater weighting on visual input? What could cause this? Is there something about the task that would make patients with CRPS (and fearful of coming into contact with anything) pay such close attention to their hand that reliance on visual input was increased? Their hands were in a ‘machine’ for starters. I wonder whether future studies could measure anxiety DURING the procedure. Or does anxiety disrupt integration in some way, helping to explain why patients lose connection with their affected hand?

I can’t help noticing that error for the unaffected hand (CRPS) in the Hidden condition seems to be larger than the affected. Is this in an interaction somewhere? If so, what is the explanation? I feel really mean, pointing this out because it’s probably nothing and not predicted at all (so why would you even look at it, right?), but it is there.

In a task that works through normal sensory integration processes, what is good or accurate performance? In a sense, being more ‘accurate’ could suggest poorer healthy integration. And this changes across time points. At T1, locating the hand close to the visual representation would suggest poor integration (total visual dominance); locating the hand close to the real hand would also suggest poor integration (total reliance on/ability to exclusively access proprioception). Somewhere in between (medium inaccurate) is good. At T6, locating at the last seen location is bad, locating at the true felt location is really good, locating somewhere in between is either ok or not being able to access proprioception. And the rate of change between T1 and T6 would tell another story. Anyway, just be careful about how you think about and describe performance and accuracy.

Things that are easy to fix:

Line 113: “importance of testing performance during implicit tasks is quite well established in studies with spatial people with neglect” In people with spatial neglect?

In parts of the paper (e.g. abstract) patients are referred to as having hand pain, but in the methods they are described as having upper limb pain. Can we have more details or more accurate descriptions of where the pain was located in the two patient groups?

When introducing the DHT, I think that it would be useful for the reader to explain that this leads to a localisation weighting that is biased towards the visual representation of the hand. This would help when, later, discussing re-weighting towards proprioception over time.

DHT procedure. If the participants move their hands outwards, why are they not aware of this? Ah, this is covered later in the manipulation check. Excellent. I wonder if it would be worth referring to, or signposting, this earlier.

Why is it called the Hidden Hand condition? When/how was the hand hidden?
Which hand were they asked to look at? How did it disappear?
It seems that the arrow always moved outwards. Would this bias localisation? In the Disappeared condition, the arrow would always pass over the real location of the hand, yes? Would this draw attention towards the real hand and/or boost the signal for its location due to congruent hand position and gaze direction? It actually makes little difference because it was the same for all groups, but in terms of future experimental robustness, I think that the arrow should start at different sides of the hand on each trial.

The methods seems to suggest that the hands always stayed in their own hemispace. Could a future manipulation be to have the hands appear to be in one hemispace, but actually be in another? (You can have that one for free!).

I think you need to explain why the error was measured in pixels. Presumably, this is because both the arrow and the hand were seen via a screen, but it can be confusing if you use terms like measuring the real position of the hand (end of self-localisation section).

What are T1-T7? These do not seem to have been mentioned before data analysis. Oh, right! These are the 6 arrow trials (perhaps call these Tx-Ty at that point), but where does the 7th come from?

It is not clear why Error over Time (T1-T7) was a factor for one hypothesis and average Error over the same time period was used for another hypothesis.

Some of the graphs in the supplementary materials refer to DHT, others to Incongruent. Please check all instances and be consistent. Although the DHT was used, incongruent (for DHT) and congruent (for Hidden) seem to make more sense here – although Static was also congruent…

Figure captions: These are not as helpful as they could be. Fig 1 – what are the individual dots? Are these individual participant scores? If so, why are there more dots than participants? Why can I not reconcile the 150+ blue dots in the CRPS group with the individual Hidden hand plots in Supplementary Figure 1? I don’t know what Figure 2 is showing. More explanation, please.

Anxiety AND CRPS seems to blow hand localisation. How interesting! I would consider changing the groupings for Fig 3. Supplemental figure 3 is much more compelling than the ms figure 3.

The first line of the discussion reads:
“This study aimed to evaluated hand-localisation …”
It either aimed to evaluate or it evaluated. I think you did evaluate, so I’d be happy with the latter, but I know that some people can get funny about that kind of thing.

The discussion claims that “people living with non-CRPS hand pain not only showed smaller difference between the DHT and control conditions – possibly indicating a smaller susceptibility to the illusion…” but this is not borne out by the statistics as far as I can see.

The tactile paragraph seems out of place here. How does it link to this experiment? Does it need to be in here at all?

---

## Round 0.2 · accepted · Accept

I think you did a very good job in revising the manuscript according to the constructive suggestions and observations made by the Reviewers. I am pretty confident that the paper will receive the deserved attention.

Reviewer 2 ·

Basic reporting

Very good

Experimental design

Very good

Validity of the findings

Very good

Additional comments

Thank you for considering all of the points raised and giving a thoughtful and balanced response to each. I look forward to seeing this published.